# Traveling-wave antenna model for terahertz radiation from laser-plasma interactions

Jiayu Zhao[1], Qining Wang[1], Yuchen Hui[1], Yamin Chen[1], Feifan Zhu[1], Zuanming Jin[1], Alexander P. Shkurinov[3], Yan Peng[1,2][⋆], Yiming Zhu[1,2][†], Songlin Zhuang[1] and Li Lao[4]

**1** Terahertz Technology Innovation Research Institute, Shanghai Key Laboratory of Modern Optical Systems, School of Optical-Electrical and Computer Engineering, University of Shanghai for Science and Technology, Shanghai 200093, China.
**2** Shanghai Institute of Intelligent Science and Technology, Tongji University, Shanghai 200092, China.
**3** Faculty of Physics and International Laser Center, Lomonosov Moscow State University, Moscow 119991, Russia.
**4** Tera Aurora Electro-optics Technology Co., Ltd, Shanghai 200093, China.

⋆ py@usst.edu.cn,    † ymzhu@usst.edu.cn

## Abstract

Generation of terahertz (THz) wave from air plasma induced by femtosecond laser pulses with a single central frequency (the so-called "single-color") is one of the fundamental interactions between light and matter, and is also the basis of subsequent pumping schemes using two- or multi-color laser fields. Recently, more states of media beyond gas (e.g., atomic cluster and liquid) via photo-ionization have brought new experimental observations of THz radiation, which can no longer be simply attributed to the mainstream model of the transition-Cherenkov radiation (TCR), thus making the whole picture unclear. Here, we revisited the mechanism of this dynamic process in a new view of the traveling-wave antenna (TWA) model. By successfully reproducing the reported far-field THz radiation profiles from various plasma filament arrangements, the wide applicability of the TWA theory has been revealed. On the other hand in the microscopic view, we investigated the plasma oscillation during filamentation aiming at further bridging the plasma filament and the antenna. Accordingly, THz-plasma resonance has been theoretically and experimentally demonstrated as the elementary THz emitter, paving the way towards fully understanding this important single-color plasma based THz source.

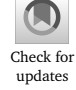

# 1  Introduction

The well-known Cherenkov radiation [1, 2] describes notable properties of radiation which appears during the motion of electrically charged particles through a substance with a velocity exceeding that of light in the same medium. In practice, the speed of the energetic charge doesn't have to be considerably superluminal since its moving trajectory normally has a finite length [3]. In this situation, boundary effects of the length-limited path also contribute to the observed radiation (i.e., the particle is assumed to gain accelerated velocities at both ending points of the trajectory) [4], and thus the Cherenkov radiation holds some features of the transition radiation [4,5] or bremsstrahlung radiation [4]. For the above reasons, the concept of transition-Cherenkov radiation (TCR) was proposed by authors of Ref. [6,7] to describe the terahertz (THz) emission from dipole-like charges moving behind the laser ionization front (at the velocity of $c$ [6]) during single-color femtosecond laser filamentation process. Finally, the TCR theory turns into the modern view for interpreting the underlying mechanism of THz

wave generation from a single-color filament [8–13], which is another milestone ever since the first demonstration in 1993 [14] of this fundamental interaction between light (intense laser) and matter (air).

However, the literal meaning of "transition-Cherenkov" is likely misleading, which cannot be understood as a real mixture of the two physical processes [15, 16], since no "transition" of particles at boundaries between two media was actually involved in Ref. [6, 7]. On the other hand, as for "Cherenkov", there is in fact few strong evidence for the speed of pumping laser dramatically exceeding that of the radiated THz wave, and this superluminality of the laser propagation has also been clarified to be not strictly necessary [3]. Furthermore, the recent hot topic of laser-induced plasma inside a liquid medium for THz wave generation [17], and the backward THz radiation featured interactions between laser and gas cluster [18], have started to fall out of scope of the TCR mechanism. Thus a more applicable theoretical frame remains as the challenge.

Remind ourselves that the TCR model is not the only option in order to account for the characteristic conical forward THz emission from a single-color filament. Meanwhile in the microwave community, another classical model of traveling-wave antenna (TWA) [19, 20] can also give birth to THz conical radiations in many flexible patterns. More importantly, the TWA model shares similar properties with the TCR theory, that is the moving point source (in TWA case, it is the current element) with a finite propagation path (in TWA case, it is the antenna itself) and accelerated velocities at both boundaries of the path [21]. Moreover, the TWA model presents its three building blocks (i.e., the element, array and space factors) in quite a concise manner as can be seen in Table 1-4, thus offering a new picture for understanding the THz wave generation from laser filaments. It is also worth mentioning that, we are delighted to see a similar thought with this work in a very recent publication [22], which has also treated the laser filaments as pulsed antennas for interpreting the THz-microwave radiation. Hence, this novel antenna's point of view is just being unfolded with many follow-up works to be deeply explored.

In the current paper, the TWA model was re-worked and improved to analytically investigate the emission properties of THz wave during single-color filamentation. Briefly, in the following Section 2, far-field profiles of THz radiation from different plasma sources were studied via the TWA method, such as a single filament without or with a longitudinal/transverse static electric field, double filaments in Vee or parallel arrangement, a micro-filament induced by laser-atomic cluster interaction, and a micro-filament excited in the water film. We have demonstrated good agreements between our theoretical results and a number of experimental observations from previous publications [6, 8–12, 17, 18]. While in Section 3, the TWA model was extended in a microscopic point of view. In short, the plasma electron oscillations during filamentation were illustrated to play the role of current elements of the TWA model, and the resonance radiation of plasma oscillators was revealed to account for THz wave generation. At last, this resonance effect has been verified by experimentally exploring its accompanying phenomenon, which is the temporal advance of THz waveforms in a bandwidth-limited detection system.

It is worth noting that, with plenty of experimental facts in Table 1-4 (including distinctly different photo-ionization media and THz radiation profiles) unified in the framework of the TWA theory, we may contribute to the recent tendency of unifying different mechanisms in a specific filamentation process. For our work, it is totally associated with the single-color laser pumping THz wave generation scheme. In other situations, e.g., in field of two-color laser excitation, four-wave mixing (4WM) and transient photocurrent (PC) models were successfully combined by building their intrinsic connections during photo-ionization of air [23–25]. Recently, a unified model was even reported [26] for both one- and two-color laser induced THz waves from a liquid, which are believed to be based on completely different mechanisms

and have usually been considered separately in the previous literature. These works offer new ideas to the community for better interpreting the THz physics in a complex laser filamentation phenomenon.

## 2 Far-field THz radiation profiles from different TWA-like plasma filament arrangements

Linear antennas, whose current distributions can be represented by traveling waves in the same direction, are referred to as traveling-wave antennas (TWA). And the corresponding TWA model describes electromagnetic (EM) radiations from this antenna, which can be written as [19, 22, 27]

$$E(\omega, \theta, l) = \frac{\eta}{2\pi r} \cdot j_0(\omega) \sin \theta \cdot Q(\omega, \theta, l), \tag{1}$$

where $\eta$ is the intrinsic impedance, $r$ and $\theta$ are for the polar coordinates with $\theta$ defined as the angle relative to the current axis $(z)$, $j_0(\omega)$ is the spectral current, and $l$ is the antenna length. As mentioned in the Introduction, the TWA model presents its formulas with three building blocks, namely, the element, array and space factors. In view of Eq. (1), the element factor (current element) is written as $j_0(\omega)\sin\theta$, which indicates the type of current and its direction of flow. In case of different plasma filament configurations, this element factor changes which can be seen in Table 1-4. As for the array factor, it will be introduced when the filaments array is studied in Section 2.5. And the space factor is Q($\omega,\theta,l$), which provides an accurate description of the phase matching condition as discussed below.

In case of THz wave emissions from a plasma-filament-based antenna, Q($\omega,\theta,l$) = $\sin[(kl/2)(K-\cos\theta)]/(K-\cos\theta)$ with $k = \omega/c$ and $K = v_{THz}/v_{g,laser}$ where $v_{THz} = c/n_{THz}$ is the THz phase velocity at $\omega$, and $v_{g,laser} = c/n_{g,laser}$ is the group velocity of the driven laser, as well as the speed of the ionization front and the traveling-wave current [22]. Normally, $K$ $(= n_{g,laser}/n_{THz}) \leq 1$ needs to be fulfilled in order to achieve the Cherenkov-type phase matching between the traveling-wave current (also the pumping laser pulse) and the field that this current radiates [22]. However, the superluminality of the laser propagation is not strictly necessary, as in the Sprangle model [3], and the THz source is generated even if the velocity of the laser ionization front is exactly the light velocity [6,7], because the THz wave can be produced due to the very finite length of the filament ($l\sim$10 mm) [7]. Thus for convenience of the following calculations, $n_{g,laser} \approx n_{THz}$ is considered, i.e., $K$ is set as 1. Hence the final expression of the radiated $\theta$-dependent distribution at a certain $\omega$ is

$$|E(\omega, \theta, l)| \propto |\sin \theta| \cdot |\frac{\sin[(kl/2)(1-\cos\theta)]}{1-\cos\theta}|. \tag{2}$$

In this case, radiations in the forward direction, i.e., along the $z$ axis, are strongly suppressed by the $|\sin\theta|$ factor [22], which leads to the typical off-axis conical THz profile, as can be seen in Table 1-4.

### 2.1 A single filament

The element and space factors of a single filament are listed in the top row of Table 1. In the same row, the schematic of a current element is also shown (as the horizontal double-headed arrow) whose oscillation direction is parallel to the $z$ axis of the filament (as the dashed line), together with its radiation profile ("8"-shaped curves) involving off-axis maximums given by $|\sin\theta|$.

Corresponding calculations were carried out at 0.1 THz for three filament lengths of $l = 10$, 30 and 100 mm. One can see that the resulted THz emission is typically in a conical shape due to the on-axis non-radiation characteristic of current elements $|\sin\theta|$. Moreover, a larger cone angle is obtained in case of a shorter filament, which can be well verified by the experimental patterns in Table 1 adapted from Ref. [6]. If the filament length is further decreased to a sub-millimeter size, the radiation pattern of the filament will be approximately like that of a current element in "8" shape. This is exactly what can be found in Ref. [28] that the THz radiation direction was nearly orthogonal to the $z$ axis. Next, we improved the TWA model and examined it in more different situations.

## 2.2 A filament with a longitudinal DC field

When the plasma filament is applied with an external DC electric field in the longitudinal direction, i.e., $E_{ex}^l$ of the order of magnitude of kV/cm or larger, the oscillation of current element will be greatly intensified (details in the Appendix A-B). Thus a coefficient $n = 1 + |E_{ex}^l/E_\omega|$ is introduced to quantitatively characterize this enhancement, as shown in the second row of Table 1, where $E_\omega \sim 0.2$ kV/cm [7] is the laser wakefield amplitude.

Then, the 0.1 THz radiation pattern produced from a 10-mm-length filament has been calculated with $E_{ex}^l = 0$ , 5 and 10 kV/cm, as black, blue and red lines, respectively. For comparison purposes, the three patterns are normalized to that of $E_{ex}^l = 10$ kV/cm. Compared with the experimental results in the same row from Ref. [8], one can observe the same phenomenon that the radiated THz intensity is increased by the longitudinal electric field, while its angular distribution remains the same.

## 2.3 A filament with a transverse DC field

THz wave radiation from a plasma filament under transverse electric fields $E_{ex}^t$ is also taken into account. This time, the THz driven current is additionally combined with a transverse current besides the original longitudinal one (details in the Appendix C). A quantitative description of this effect via the TWA theory can be that the direction of the current element is rotated by a certain angle $\varphi = \arctan(E_{ex}^t/E_\omega)$ as shown in the third row of Table 1. In this condition, the symmetry of the 0.1 THz radiation pattern from a filament ($l = 20$ mm) is broken. With a larger bias voltage (i.e., a larger $\varphi$), the convergence of two lobes develops towards the $z$ axis and keeps asymmetrical all the time. This external field effect has also been observed in Ref. [9] as shown as the experimental results in the same row.

When the bias voltage reaches saturation values making $\varphi$ into 90°, the current element will oscillate perpendicularly to the $z$ axis, as shown in the fourth row. Under this circumstance, the produced 0.1 THz radiation after the laser filament ($l = 15$, 50 and 100 mm) is on-axis restrained in a smaller angular distribution for a longer filament. This well agrees with the experimental results in Ref. [9].

## 2.4 Vee bi-filaments

In this section, we studied the THz radiation from a Vee-antenna constituted by dual-filaments with a crossed half angle $\alpha$ , as shown in the Row 1 of Table 2. Here, $\theta$ in $E(\omega, \theta, l)$ is replaced by $(\theta + \alpha)$ for the upper filament, or by $(\theta - \alpha)$ for the lower one. Then, two situations are considered: THz waves from the two filaments are (i) in phase and expressed as $|E(\omega, (\theta + \alpha), l) + E(\omega, (\theta - \alpha), l)|$; (ii) out of phase as $|E(\omega, (\theta + \alpha), l) - E(\omega, (\theta - \alpha), l)|$. Meanwhile, $\alpha$ and $l$ were set to be 32° and 10 mm, respectively.

The calculation outcomes for emission profiles at 0.1 THz from this V-shaped filaments array are displayed in Row 1 of Table 2. For case (i), THz waves from the two filaments

achieve destructive interference and cancellation in the $z$ direction. While for case (ii), the THz intensity is enhanced by $\sim$ 4 times along the $z$ axis due to constructive interferences. These two patterns well match the experimental results in Ref. [10] as shown in the right side of the same row. It is worth mentioning that, our results are even closer to these experimental observations than the theoretical results in Ref. [10] given by the TCR model.

## 2.5 Parallel bi-filaments

In another condition of a plasma array being formed by two parallel filaments, the entire THz wave emission is the superimposition of THz radiations from parallel-oscillated current elements along both filaments (in Table 2). In our model, the array factor $|\cos(kd\sin\theta/2-\beta/2)|$ is introduced, where $d$ and $\beta$ denote the spatial separation and relative temporal (phase) delay between two parallel filaments, respectively. Now, the total THz radiation of the bi-filaments is the product of element, array and space factors.

We first varied $d$ (= 0.5, 2.4 and 3.4 mm) with a fixed $\beta$(= 0) and studied the synthesized 0.1 THz intensity profile from 10-mm-long bi-filaments. The corresponding three calculation results are shown in the second row of Table 2. It can be seen that with the increasing of $d$, the THz radiation diminishes but remains symmetric. Similar experimental phenomena were also observed in Ref. [11].

Next, we considered different $\beta(0-2\pi)$ between THz pulses from the laser filaments with a fixed $d$ at 2.4 mm. This time, the THz radiation pattern undergoes asymmetrical changes as recorded in the third to fifth rows of Table 2. Briefly, to begin with $\beta = 0$, the THz emission is symmetric. With the growth of $\beta$, the radiation pattern becomes asymmetric. When $\beta$ equals $\pi/4$ or $7\pi/4$, the THz angular profile is completely in the direction of one single lobe. With further delay (after $\beta = \pi$), the other lobe starts to rise at the sacrifice of the former. At $\beta = 2\pi$, the radiated THz pattern is identical to the one without time/phase delay (i.e., the first figure in Row 3). Finally, good agreements between our calculations and experimental reports in Ref. [11] can be clearly seen in Table 2.

## 2.6 Parallel bi-filaments with a transverse DC field

In Table 3, on the basis of the above parallel bi-filaments, an additional transverse DC field was applied. In this case, the element factor $|\sin\theta|$ has been turned into $|\cos\theta|$ according to the last row of Table 1. Meanwhile, the other factors are unchanged. Afterwards, the same calculation processes as the ordinary parallel bi-filaments (in Table 2) have been repeated and the results are presented in the top four rows of Table 3. In Row 1, for a larger distance $d$ between two filaments ($\beta = 0$), the THz emission pattern becomes less opening, but remains to be symmetrical around the $z$ axis. In Row 2-4, for a varied $\beta$ ($0-2\pi$) and a fixed $d$ (2 mm) between the double filaments, the resulted THz emission pattern gradually turns to be asymmetrical, alternatively oriented to one side of the $z$ axis. These characteristic evolutions of THz far-field profiles have been well reproduced in reported experiments [12].

## 2.7 A filament with transverse jet of atomic cluster

In this section, the THz radiation from atomic cluster plasma was investigated. As for this scheme, electrons in the plasma column are partly driven away from the laser propagation direction, which accordingly generates a transverse current element, preferring to be parallel to the direction of gas injection [18]. Combined with the original current element along the $z$ axis, a quasi-quadrupole is created for the element factor which can be described as $|\sin\theta \times \cos\theta|$. A typical angular distribution of THz radiation from this orthogonal current element

inside the plasma cluster was calculated at 0.1 THz and shown in Row 5 of Table 3, in which it can be seen that four lobes are in about ±45° with respect to the laser propagation axis.

When the filament length $l$ is increased from 0.7 to 1.8 mm, the whole THz emission pattern keeps symmetrical and the backward lobes gradually fade away. This property of backward THz emission from a short plasma filament induced by laser-cluster interaction has also been experimentally confirmed in Ref. [18] as shown in the same row. The remaining discrepancy of the angle of main lobes between our calculated patterns (±45°) and the reported observation (±30°) might be attributed to the fact that, a broadband THz signal was measured by the used bolometer in the literature [18], while our results were calculated at a single THz frequency. Moreover, this backward THz emission phenomenon cannot be easily explained by the popular Cherenkov-type mechanism.

## 2.8 A filament excited in the water film

In recent years, great efforts have been made on new states of matter (beyond gas and solid) being photo-ionized for THz wave generation, such as liquid THz sources. Water films [17, 29–37] in both flowing and static forms were studied, as well as the liquid nitrogen [38,39] or even the liquid metal [40,41]. The generated THz energy was proved to be >10 times larger than that achieved from the most standard table-top technique, i.e., the two-color filament in air [37]. Here in this section, we show that this new liquid-plasma target, inside which THz wave radiates, also follows the basic nature of the improved TWA theory.

Taking a water film as an example which is drawn in light blue color in Table 4, the incident laser beam produces two angles of $\alpha$ and $\theta_r$ when it is refracted on the interface between air and water. Inside the water film, a micrometer-length filament is created at the laser focus and radiates THz wave, which then makes angles of $\theta_t$ and $|\beta - \alpha|$ during propagation through the other water-air interface towards the detection point (marked with an eye). Here, $|\beta - \alpha|$ denotes the detection direction with respect to the horizontal $z$ axis, where $\beta$ is the included angle between directions of the detection and the initial laser propagation. Note that, the minus (-) is written due to the opposite signs between $\beta$ and $\alpha$. Furthermore, $\theta_r = \arcsin[\sin(-\alpha)/n_{laser}]$ and $\theta_t = \arcsin[\sin(\beta-\alpha)/n_{THz}]$ according to the Snell's Law applied on both surfaces of the water film. $n_{THz} = 2.33$ and $n_{laser} = 1.33$ [42] are the refractive indices of THz wave and laser inside the water, respectively.

As for the above filament, its THz emission pattern can be easily described by replacing $\theta$ in $E(\omega, \theta, l)$ with $(\theta_t + \theta_r)$. At the same time, the laser energy loss caused by its reflection on the air-water interface is also taken into account by introducing the attenuation factor of $(1-r^2)$, where r follows the Fresnel Equation expressed as $r = (\cos\alpha \text{-} n_{laser}\cos\theta_r)/(\cos\alpha + n_{laser} \times \cos\theta_r)$. On the other hand, however, the THz energy loss due to the water absorption is not considered for the following reasons. According to Ref. [17], the THz intensity through the water film is given by $I_{THz} = I_0 \exp[\text{-}\alpha' d/(2\cos\theta_t)]$, where $\alpha' = 100$ cm$^{-1}$ is the power absorption coefficient, and $d = 120$ $\mu$m is the thickness of the water film. On the water-to-air interface, THz waves can't be coupled out if its total internal reflection occurs, and the critical angle can be easily calculated as 25 degree. Thus $\theta_t$ varies between -25 and +25 degree, and $I_{THz}$ is between $0.51I_0$ and $0.54I_0$, where $\Delta = 0.03I_0$ is negligible when profiling the angular distribution of the THz radiation. At last, calculations for the THz wave generation (centered at $f = 0.5$ THz) from a micro-plasma filament ($l = 40$ $\mu$m) were performed by multiplying the two factors of element and space in Table 4 with the attenuation factor.

The resultant THz intensities $I_{THz}$ are plotted as a function of $\alpha$ in Table 4 (as red lines) with $\beta$ fixed at 0° or 55°, providing good reproductions of the reported results in experiments [17]. On the other hand, $I_{THz}(\alpha, \beta)$ was also calculated and shown in Table 4 as the 2-D figure. Compared with the reported one on the right-hand side, the overall trends of $I_{THz}$ evolution along $\alpha$ and $\beta$ are exactly the same. That is, with the increase of $\beta$, the $I_{THz}$ dip gradually shifts

towards a greater $\alpha$. Note that contradictions appear near the white dashed lines (given by $|\beta - \alpha| = 90°$) in the two figures, where $I_{\text{THz}}$ achieved minimum in the publication. This point is easily understood since $I_{\text{THz}}$ cannot be observed in the direction orthogonal to the horizontal $z$ axis (i.e., $|\beta - \alpha| = 90°$), where the water film either blocks the detector or seriously reduces the THz energy. For the sake of convenience, this issue is neglectable in our model.

**Table 1-4. Formulas for angular distributions of THz emission given by the improved traveling-wave antenna (TWA) model, based on which calculations have been carried out and the results are compared with the reported experimental ones.** In the tables, different types of THz sources were taken into account, including a single filament without or with a longitudinal/transverse static electric field, bi-filaments in Vee or parallel shape (without or with a transverse DC field), a micro-filament induced by laser-gas cluster interaction, and a micro-filament created in the water film. Note that, EF in Table 2-4 is short for "element factor". The reused figures of experimental results adapted from Ref. [6, 9] are with permissions of the corresponding author and American Physical Society; Figures adapted from Ref. [8, 11, 17, 18] are with permissions of AIP Publishing; Figures adapted from Ref. [10] are with permissions of IOP Publishing; Figures adapted from Ref. [12] are with permissions of Springer Publishing.

Table 1

| The re-worked traveling-wave antenna (TWA) model | | | Our calculations | | | | | Reported experiments | | | |
|---|---|---|---|---|---|---|---|---|---|---|---|
| Type | Element factor | Space factor | Current element | $l=10$ mm | 30 mm | 100 mm | | $\sim 10$ mm | $\sim 30$ mm | 80 mm | [6] |
| Single filament | $|\sin\theta|$ | | | | | | | | | | |
| Single filament with a longitudinal DC field | $n|\sin\theta|$ | $\left|\dfrac{\sin[\frac{kl}{2}(1-\cos\theta)]}{1-\cos\theta}\right|$ | | 0 kV/cm | 5 kV/cm | 10 kV/cm | | 0 (black), 5 (blue) and 10 (red) kV/cm | | | [8] |
| Single filament with a transverse DC field | $|\sin(\theta+\varphi)|$ or $|\cos\theta|(\varphi=90°)$ | | | 0 kV/cm | 0.1 kV/cm | 1 kV/cm | | 0.5 kV | 3 kV | | [9] |
| | | | | $l=15$ mm | 50 mm | 100 mm | | 15 mm | 50 mm | | |

Table 2

| | The re-worked TWA model | | Our calculations | | | Reported experiments | | |
|---|---|---|---|---|---|---|---|---|
| Type | Element factor | Space factor | Current element | In phase | Out of phase | | In phase | Out of phase |
| Vee bi-filaments | $\|\sin(\theta \pm \alpha)\|$ | $\left\|\dfrac{\sin\left\{\frac{kl}{2}[1-\cos(\theta\pm\alpha)]\right\}}{1-\cos(\theta\pm\alpha)}\right\|$ | | $d=0.5$ mm | $2.4$ mm | $3.4$ mm | $0.5$ mm | $2.4$ mm / $3.4$ mm |
| Type | EF | Array factor | Space factor | Current element | | | In phase | Out of phase |
| Parallel bi-filaments | $\|\sin\theta\|$ | $\left\|\cos\left(\dfrac{kd}{2}\sin\theta - \dfrac{\beta}{2}\right)\right\|$ | $\left\|\dfrac{\sin\left[\frac{kl}{2}(1-\cos\theta)\right]}{1-\cos\theta}\right\|$ | | $\beta=0$ / $\pi/4$ / $\pi/2$ | | $0$ ps / $1.7$ ps / $3.3$ ps | |
| | | | | | $3\pi/4$ / $\pi$ / $5\pi/4$ | | $3.7$ ps / $5.3$ ps / $7.3$ ps | |
| | | | | | $3\pi/2$ / $7\pi/4$ / $2\pi$ | | $9$ ps / $10.3$ ps | |

[10]   [11]

Table 3

| The re-worked TWA model | | | | Our calculations | | | | Reported experiments | | |
|---|---|---|---|---|---|---|---|---|---|---|
| Type | EF | Array factor | Space factor | Current element | $d=1$ mm | 2 mm | 5 mm | $d=1$ mm | 2 mm | 5 mm |
| Parallel bi-filaments with a transverse DC field | | $\left|\cos\theta\right|\left|\cos\left(\dfrac{kd}{2}\sin\theta-\dfrac{\beta}{2}\right)\right|$ | $\left|\dfrac{\sin\left[\dfrac{kl}{2}(1-\cos\theta)\right]}{1-\cos\theta}\right|$ | | $\beta=0$ | $\pi/4$ | $\pi/2$ | 0 ps | 2 ps | 4 ps | [12] |
| | | | | | $3\pi/4$ | $\pi$ | $5\pi/4$ | 5 ps | 5.5 ps | 6 ps |
| | | | | | $3\pi/2$ | $7\pi/4$ | $2\pi$ | 7 ps | 9 ps | 11 ps |
| Type | | Element factor | | Current element | $l=0.7$ mm | 1 mm | 1.8 mm | | ~3 mm | |
| Single filament with transverse jet of atomic cluster | | $\left|\sin\theta\cdot\cos\theta\right|$ | | | | | | | | [18] |

Table 4

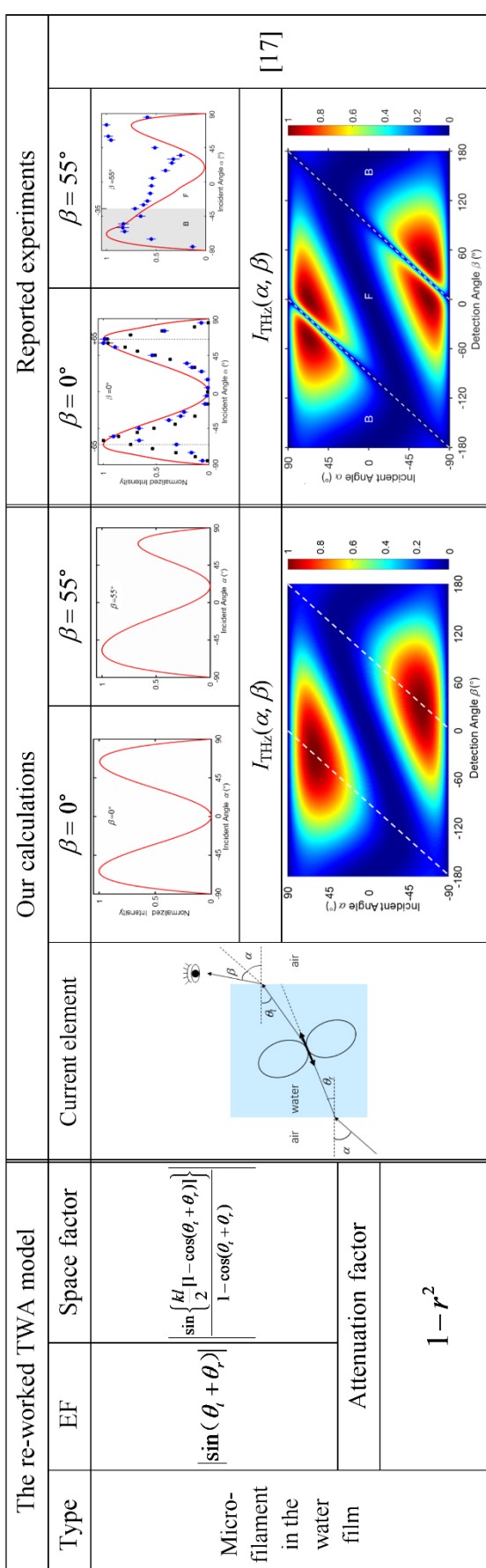

# 3 THz resonance radiation from length-controlled plasma filaments

## 3.1 The THz-plasma resonance effect and the corresponding THz spectral peak

According to Section 2, a plasma filament plays similar role of a TWA, which can be treated as a line source with periodic basic current elements. And the total field emission from the antenna is the superposition of waves radiated from individual current element emitters [21]. By this means, the forward radiation profiles of THz wave have been well interpreted as shown in Table 1-4. This, however, seems to mainly solve the THz intensity distributions in the far field, without considering enough details associated with the near-field element emitter. In Section 2.2 and 2.3, we briefly mentioned Appendix A-C by introducing the gain coefficient $n$ and the rotation angle factor $\varphi$ to the TWA model. These three Appendix sections in fact treated the plasma filament in the microscopic view, thus playing the role of bridging the macroscopic TWA model and the microscopic model.

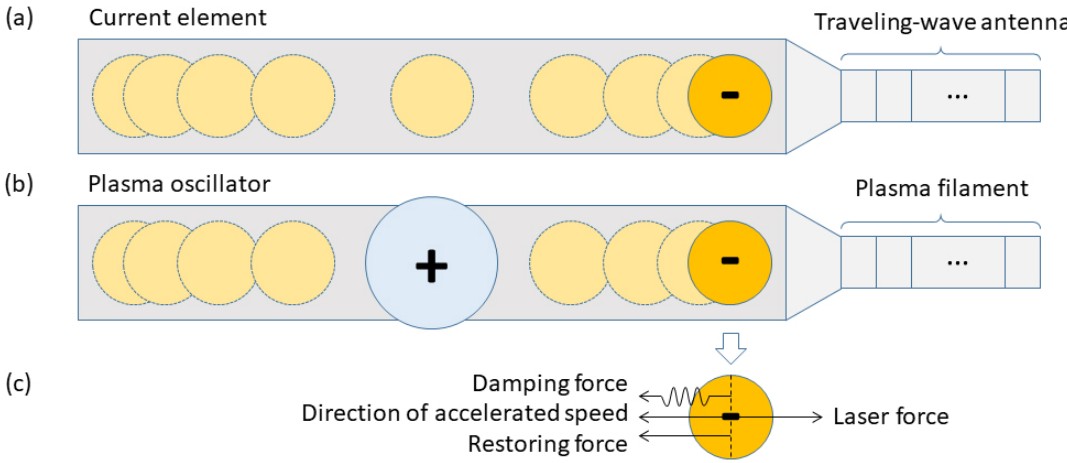

Figure 1: **Microscopic view between the traveling-wave antenna and the plasma filament.** (a) The schematic of a traveling-wave antenna with periodic current elements, inside which the electron (-) gains accelerated velocities at both terminals. (b) Similar case with a plasma filament. At this time, the stationary positive ion (+) is contained inside the elementary plasma oscillator. (c) The force analysis of an electron during plasma oscillation. When the "Restoring force" is solely responsible for driving the "accelerated speed", the "Laser force" energy is then totally converted into "Damping" in form of THz radiation, which then achieves maximum.

In microscopic point of view during filamentation, plasma electron oscillations along the plasma filament actually share one physical picture with the current element inside the TWA, e.g., at the beginning and ending points of each plasma dipole oscillation or current element, the moving electron experiencing acceleration and deceleration, respectively [21], as schematically shown in Fig. 1(a-b).

Specifically, the process of plasma oscillation can be described as an electron being taken away from the equilibrium position in the post-ionization stage, and the final radiated spectrum is given by (Appendix A)

$$E_{rad}(\omega) \propto i\omega\varepsilon_0 E_L \omega_{pe} \frac{\omega + 2i\nu_e}{2(\omega^2 - \omega_{pe}^2 + i\nu_e\omega)} \times \frac{\sin(\omega\tau_L/2)}{1 - (\omega\tau_L/2\pi)^2} e^{-i\omega\tau_L/2}, \tag{3}$$

where $E_L = e\omega_{pe}I_0/2m_e\varepsilon_0 c^2\omega_0^2$ is the amplitude of the plasma wave driven by the laser pon-

deromotive force, $\omega_{pe} = (N_e e^2/m_e \varepsilon_0)^{1/2}$ is the plasma frequency, $\nu_e$ is the electron collision frequency and $\tau_L$ is the laser pulse duration. One may notice that the first half of Eq. (3) (before the multiplication sign) mainly includes parameters relevant to the plasma such as $\omega_{pe}$ and $\nu_e$, while the second half only involves characteristics of the laser field, e.g., $\tau_L$. Hence, Eq. (3) is divided into two parts in the form of $E_{rad}(\omega) \propto E_p(\omega) \times E_l(\omega)$. As for $E_p(\omega)$, it is the spectral component dominated by the plasma parameters in the first half of Eq. (3) and can be simplified as:

$$E_p(\omega) \propto i\omega\omega_{pe}^2 \frac{\omega + 2i\nu_e}{\omega^2 - \omega_{pe}^2 + i\nu_e\omega} . \tag{4}$$

As for $E_l(\omega)$, it can be written as the following expression:

$$E_l(\omega) \propto \frac{\sin(\omega\tau_L/2)}{1 - (\omega\tau_L/2\pi)^2} e^{-i\omega\tau_L/2} . \tag{5}$$

Next, calculations of the radiated spectra based on Eq. (3-5) were performed with typical parameter values: $N_e = 1\times10^{17}$ cm$^{-3}$, $\lambda = 800$ nm, $\tau_L = 50$ fs, $I_0 = 1\times10^{14}$ W/cm$^2$ and $\nu_e$ = 2 THz. It is worth mentioning that, this $\nu_e$ value is supported by both previous publications [3,6,7,9,43,44] and our calculations (Appendix E). It is true that when gases are fully ionized [45], or in case of photo-ionization induced by CEP-stable few-cycle laser pulses [46], the electron collision can be strongly accelerated and $1/\nu_e$ will be even on timescale of 1 fs [46], resulting in $\nu_e \sim 10^3$ THz. However, this is obviously not our condition with air being weakly ionized by multi-cycle laser.

The results are presented in Fig. 2(a), in which the total THz radiation given by Eq. (3) ($E_{rad}(\omega)$ as the black solid line) is basically the combination of a low-frequency peak given by Eq. (4) ($E_p(\omega)$ as the blue dotted line) and high-frequency sinusoidal steps given by Eq. (5) ($E_l(\omega)$ as the red dashed line). This hints that the low-frequency peak of $E_{rad}(\omega)$ is mainly associated with the plasma, while the high-frequency sinusoidal steps are decided by the laser field.

The above calculated results agree with theoretical and experimental reports in the literature [13,47], especially for the sinusoidal steps. As for the spectral peak, it actually appears at the plasma frequency $\omega_{pe} = (N_e e^2/m_e \varepsilon_0)^{1/2}$ inside which $N_e$ is the only variable. If increasing $N_e$ from $3\times10^{16}$ to $12\times10^{16}$ cm$^{-3}$, one can clearly see the blue shift of the spectral peak from 1.6 to 3.2 THz in Fig. 2(b$_1$). Homogeneous plasma oscillations were suggested as the source of this spectral maximum at the plasma frequency [23,48,49]. However, these oscillations are theoretically nonradiative [3,50–57]. A question is then naturally raised: what is the origin of this $\omega_{pe}$-dependent THz yield?

By carefully reading Eq. (4), the spectral peak could be predicted from the equation's denominator when $\omega_{THz} = \omega_{pe}$, i.e., resonance happens between THz waves and plasma oscillations. In this case, the first two items in Eqs. (7,9,10) of the Appendix cancel each other, which means that the restoring force $eE$ solely drives the electron oscillation. And thus the laser energy (the fourth item in Eqs. (7,9,10)) mainly compensates the loss due to the oscillation damping (the third item), which is converted into the THz radiation burst (see also the force analysis in Fig. 1(c)). The clue of plasma-THz resonance can also be traced back to Ref. [14], which indicated that (i) the frequency of THz emission varied with the gas density (by changing the ambient gas pressure) and was close to the bulk plasma frequency as shown in its Fig. 3, and (ii) the THz signal was attributed to the nonlinear current in its Eq. (1), whose denominator contains the item of $[1-(\omega_{pe}/\omega)^2]$. Both (i) and (ii) reveal that maximum THz signals could be achieved at $\omega_{THz} = \omega_{pe}$ (i.e., resonance effect). It is worth mentioning that, here $\omega_{pe}$ is more appropriate than $\omega_{pe}/\sqrt{2}$ or $\omega_{pe}/\sqrt{3}$ for the plasma filament being studied, whose length is significantly larger than its column diameter. Furthermore, spectral

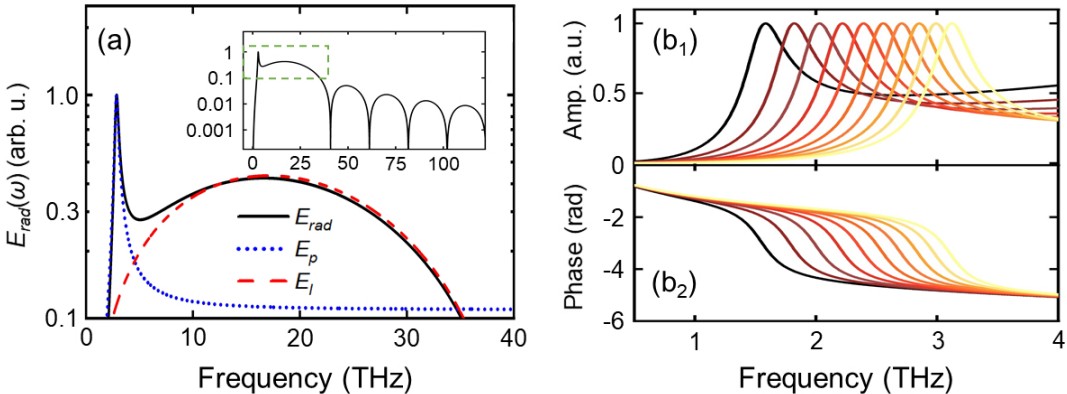

Figure 2: **The amplitude and phase spectra of THz radiation given by the resonance effect.** (a) The calculated THz amplitude radiation in frequency domain by Equation (3-5) as the total $E_{rad}(\omega)$ (black solid line), low-frequency $E_p(\omega)$ (blue dotted line) and high-frequency $E_l(\omega)$ (red dashed line), respectively. (a) is enlarged from the green dashed box in its inset, which is the full view of $E_{rad}(\omega)$ along the frequency axis. (b$_1$) The amplitude spectrum of the low-frequency peak as a function of THz frequency with an increasing $N_e$ ranging from 3, 4, 5 ... to $12{\times}10^{16}$ cm$^{-3}$ (from left to right). (b$_2$) The corresponding spectral phase variations.

phase evolutions corresponding to Fig. 2(b$_1$) are shown in Fig. 2(b$_2$). Around each peak frequency in Fig. 2(b$_1$), phase variations of $\pi$ (3.14 in rad) can be observed in Fig. 2(b$_2$), further verifying the THz-plasma resonance effect. In the following section, superpositions of this elementary resonance radiation of THz wave along a single-color filament would be discussed.

## 3.2 Indication of the resonance effect: temporally periodic oscillation and its phase variation

Along a photo-ionization induced plasma filament in air, the longitudinal plasma density $N_e$ distribution can be simulated by the nonlinear wave equation considering the slowly varying envelope approximation [58]:

$$2ik_0\frac{\partial A}{\partial z} + \Delta_\perp A + 2(1 + \frac{i}{\omega}\frac{\partial}{\partial \tau})\frac{k_0^2}{n_0}(\Delta n_{Kerr} + \Delta n_{plasma})A - k_2 k_0 \frac{\partial^2 A}{\partial t^2} - ik_0\alpha A = 0. \quad (6)$$

Eq. (6) contains several optical effects, including self-focusing, diffraction, group-velocity dispersion, and plasma generation due to multi-photon-tunnel ionization, etc. Here, $k_0$ and $k_2$ represent wave number and group velocity dispersion parameter, respectively. $\alpha$ is the absorption coefficient associated with ionization in air. And $A$ is the electric field envelope function. The following initial parameters were adopted for simulations of $N_e$: 50 fs, 800 nm, 1 kHz and 1 W laser pulses were focused by a lens with $f = 100$ cm.

The simulated on-axis distribution of $N_e$ along the filament is shown in Fig. 3(a). The maximum of $N_e$ appears at $z = 0$ mm which is defined as the focal point of the laser. Note that, the plasma could be produced within the same temporal scale of the pumping laser pulse duration, e.g., a few tens of femtoseconds [59], while the plasma's average lifetime is as long as several nanoseconds [60]. This fact hints that, the plasma could be formed in an instant moment before the THz pulse generation. Afterwards, it would last enough long time for THz-plasma interactions. For this reason, we paid little attention on the temporal evolution

of $N_e$ in the following treatments. Based on the $N_e$ evolution in Fig. 3(a) and Eq. (3) of $E_{rad}(\omega)$, 2-D THz amplitude distribution as functions of $z$ and THz frequency were calculated and displayed in Fig. 3(b). One can see that three horizontal dark-color areas within 80 THz correspond to the first three sinusoidal steps like that in the inset of Fig. 2(a). And the low-frequency component (< 4 THz) in the dashed black box is further enlarged and presented in the inset of Fig. 3(b), in which the spectral peak frequency increases first and then decreases. This peak frequency variation follows the $N_e$ evolution (in Fig. 3(a)) as previously illustrated in Fig. 2($b_1$).

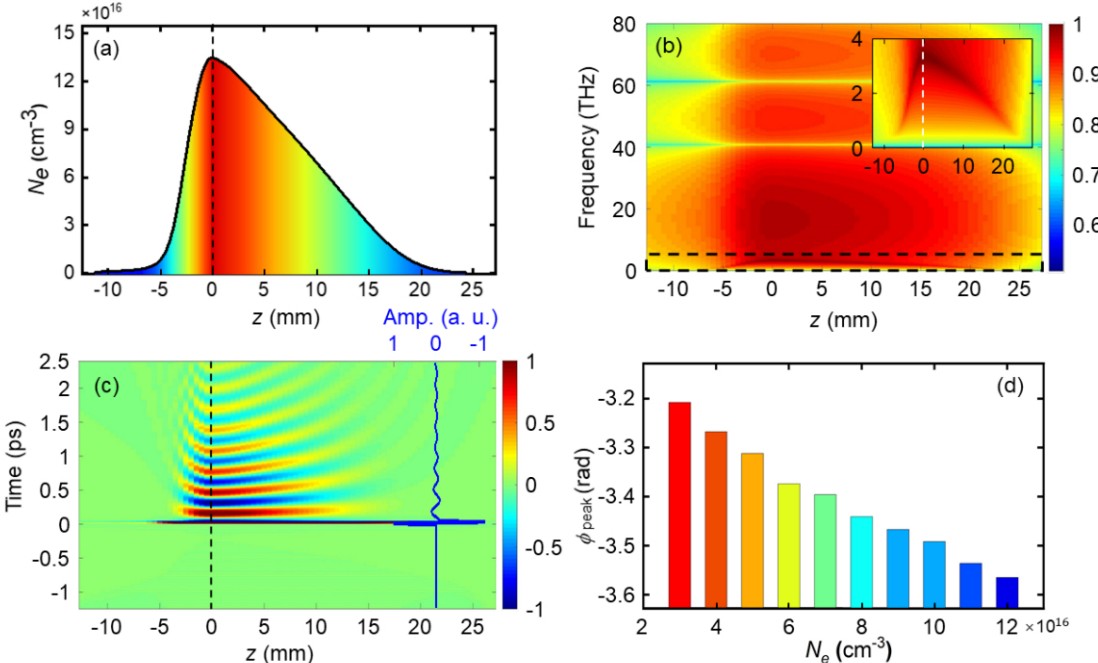

Figure 3: **THz signals in both time and frequency domains along the plasma filament with an inhomogeneous $N_e$ distribution.** (a) The simulated longitudinal Ne distribution along the propagation axis $z$ of the plasma filament. (b) 2-D THz amplitude distributions as functions of $z$ and THz frequency. Inset: the enlarged view of low THz band (<4 THz) within the black dashed box. (c) 2-D THz amplitude distributions as functions of $z$ and time delay, achieved by inverse Fourier Transformations (iFT) on (b). The blue waveform on the right-hand side, given by $z$-integration, denotes the THz transient emitted from the entire plasma filament. (d) The phase of spectral peak $\phi_{peak}$) decreases with the increasing $N_e$, obtained from Fig. 2($b_2$).

In next step, inverse Fourier Transformations (iFT) were performed on the data in Fig. 3(b), and the corresponding results are shown in Fig. 3(c) as 2-D THz temporal waveforms with respect to $z$. In the right side of Fig. 3(c), the blue THz waveform is given by $z$-integration of the 2-D data. Theoretically, sinusoidal steps and spectral peaks in Fig. 3(b) are expected to be iF-transformed into single-cycle pulses and periodic oscillations in time domain, respectively. This is actually what can be seen in Fig. 3(c): the single-cycle main pulse of THz radiation is located around $t = 0$ ps, and periodic oscillations are after the THz main pulse along the time axis.

It can also be noticed in Fig. 3(c) that 2-D "stripes" bend at the vicinity of $z = 0$ mm. Now, attentions were focused on this bending phenomenon, which in fact indicates temporal advance (before $z = 0$ mm) and delay (after $z = 0$ mm) of the periodic oscillation. In other words, the phase of temporal oscillations (at spectral peak frequency) decreased first (before $z$

= 0 mm) and then increased (after $z = 0$ mm). On the other hand, one can see in Fig. 3(a) that $N_e$ increased and decreased before and after $z = 0$ mm, respectively. Thus, the growth of $N_e$ might contribute to the phase decreasing at the spectral peak frequency, and vice versa. This hypothesis has been proved by extracting the phase value at peak frequency under different $N_e$ in Fig. 2(b$_2$) and plotting the results in Fig. 3(d), in which the spectral peak's phase is in anti-correlation relationship with $N_e$ as expected.

## 3.3 Distortion of the temporal oscillations due to the limited bandwidth of the THz detection system

The above temporally periodic oscillation after the main THz pulse, and its phase variations with respect to $N_e$, are rarely reported in previous publications. Possible reasons are discussed in this section. In experiments, the plasma filament generated by focusing femtosecond laser pulses in air usually has a plasma density $N_e$ greater than $10^{16}$ cm$^{-3}$ [61]. Therefore, the corresponding resonance frequency ($\omega_{pe}/2\pi$) is > 0.89 THz. Coincidentally, this frequency value is close to the upper cut-off frequency of the constantly used ZnTe crystal in the traditional electro-optic (EO) sampling setup for temporal THz waveforms detection. For example, an 1.5-mm-thick ZnTe crystal has an upper cut-off frequency no more than 1 THz [62], beyond which the THz spectral signal would be seriously attenuated by the material phonon absorption. Accordingly, the proposed spectral resonance peaks along the plasma filament (in Fig. 3(b)) are very likely to be erased due to this finite bandwidth of the EO crystal, especially around the laser focus where $\omega_{pe}$ is relatively larger.

Consider the response function of a ZnTe crystal as a shifted Gaussian function of $R(f) =$ exp$[-2(f\text{-}f_0)^2/\Delta f^2]$, where ($f_0$, $\Delta f$) are set to be (0.2, 1.6), (0.1, 0.8) and (0.05, 0.4) THz in the following studies. Thus the corresponding upper cutoff frequencies at $1/e^2$ of the Gaussian distributions are about 1.8, 0.9 and 0.45 THz, respectively. These response functions for low-pass filtering are shown in the left-hand side of Fig. 4(a$_1$-a$_3$) as white lines, and the main part of Fig. 4(a$_1$-a$_3$) are the filtered 2-D THz distributions achieved by multiplying the response functions with the data originally shown in the inset of Fig. 3(b). One can see in Fig. 4(a$_1$) that the spectral peaks around $z = 0$ mm have been filtered out, and this phenomenon is even more serious in Fig. 4(a$_2$-a$_3$) due to the narrowed window width. Meanwhile in time domain as shown in Fig. 4(b$_1$-b$_3$), this low-pass filtering effect mainly reduces the "stripe" bending around $z = 0$ mm, where temporal advance or delay can no longer be observed. Furthermore, the main THz single-cycle pulse and the following periodic oscillations gradually combine into a slowly-varying waveform, whose duration gets much wider towards Fig. 4(b$_3$).

Fig. 4(a$_1$-a$_3$) and 4(b$_1$-b$_3$) display the THz spectral and temporal signals generated at each $z$ position. In actual experiments, the directly detected data should be the coherent combination of pulses along the plasma filament with a certain length. Moreover, this length can be controlled by inserting a blocker (perpendicularly to the $z$ axis) inside the filament [63]. Therefore, the THz waveforms in time domain as shown in Fig. 4(b$_1$-b$_3$) were integrated as a function of $z$, and the calculated consequences are presented in Fig. 4(c$_1$-c$_3$). It can be seen that at the leading end of the filament ($l < 10$ mm), there is a temporally forward shift of THz pulses. Nevertheless, this trend is no longer evident in case of $l > 10$ mm, where the displacement of the waveform maximum tends to be a constant. THz waveforms at $l = 35$ mm as final signals radiated from the whole plasma filaments are extracted and plotted on the right-hand side of Fig. 4(c$_1$-c$_3$) as purple lines. One can clearly see the missing of temporally periodic oscillations after the main THz pulse (in comparison with Fig. 3(c)). In this situation, only the temporal advance of THz pulses at the front end of the filament could account for the occurrence of the THz-plasma resonance. This tendency has been illustrated by a white dashed curve in Fig. 4(c$_3$), which highlights the time-domain locations of the maximum THz amplitude, and would also be experimentally verified in the next section.

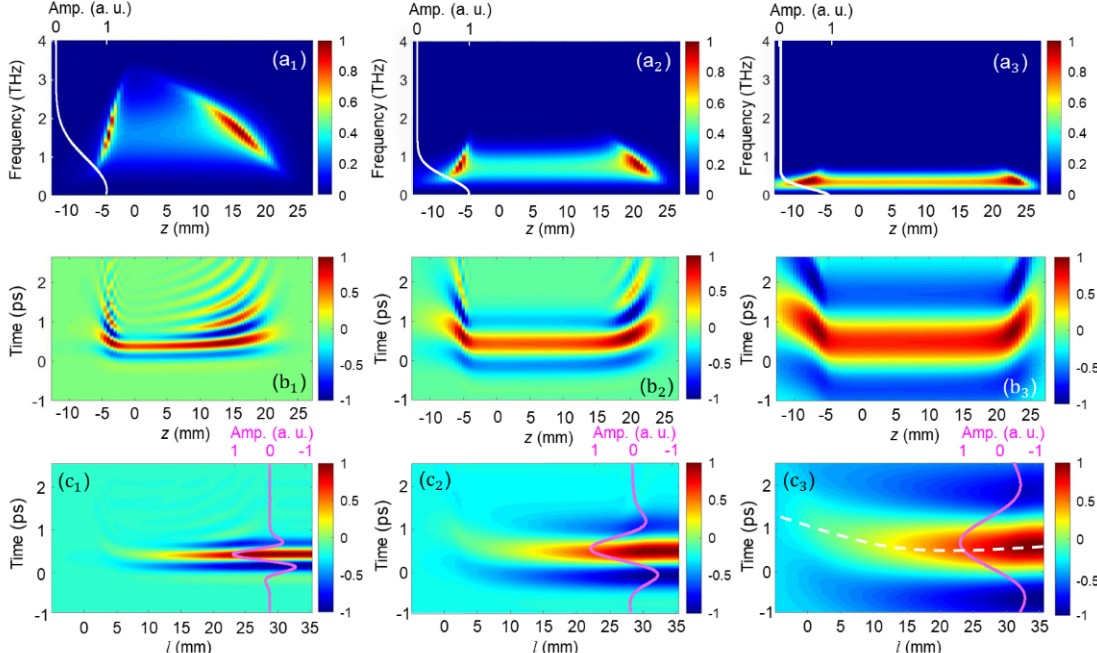

Figure 4: **The temporal THz waveforms detected with different bandwidths.** (a) The low-pass filtered results of 2-D THz spectrum distributions, achieved by multiplying the data in Fig. 3(b) by the window functions (white lines) overlapped on the left side of $(a_1$-$a_3)$, whose upper cutoff frequencies (at $1/e^2$) are 1.8, 0.9 and 0.45 THz, respectively. (b) The corresponding 2-D THz distributions in time domain at each point $z$. (c) The integrated outcomes of (b) as a function of the filament length $l$. The purple lines highlight the final THz waveforms from the whole plasma filament.

### 3.4 Experimental demonstrations of the THz-plasma resonance effect by observing the temporal advance phenomenon

In this section, experiments have been carried out in order to confirm the theoretical predictions about the temporal advance of THz waveforms in Fig. 4($c_1$-$c_3$) as the indirect evidence for the resonance behavior between THz and plasma waves. Thus, the laser equipment operated with the same parameters as described in Section 3.2 for THz wave generation during filamentation in the air. Specifically, the blocker used for varying the filament length was a ceramic plate in thickness of about 0.8 mm [64]. In addition, the detection setup adopted the typical THz-TDS scheme [65], in which a 1.5-mm-thick ZnTe crystal was adopted for EO sampling. Finally, temporal THz waveforms radiated from length-controlled filaments were detected.

The experimental results are shown in Fig. 5(a), whose differential outcomes are indicated in Fig. 5(b). Both figures well reproduced the computed temporal shift of the single-cycle THz pulse as displayed in Fig. 4($c_3$) and (b$_3$), respectively. Fig. 5(c) further quantified the displacements of the THz waveform maximums along the dashed white lines in Fig. 5(a) and Fig. 4($c_3$), as red circles and black line, respectively, and their fitting with each other is acceptable. Note that in Fig. 5(c) the red open circles have been shifted as a whole along the ps-axis (for better comparison with the black line), which is reasonable since the temporal zero point of the detected THz waveform via the pump-probe method is arbitrary.

It is also worthy of mentioning that, our scheme of recording the temporally advanced THz waveforms (in Fig. 5(a)) is a compromise since few candidates are available. One may

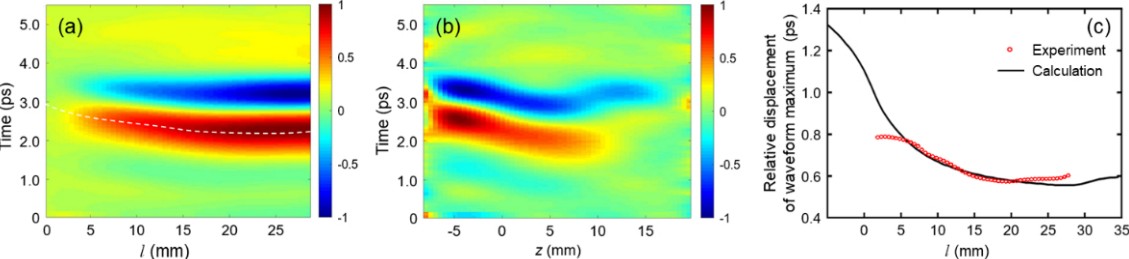

Figure 5: **Comparison between the experimental and theoretical results of THz wave temporal shift.** (a) The recorded THz waveforms emitted from different filament lengths $l$. (b) THz pulses generated at each $z$ position, achieved by applying differential on the data in (a). (c) The relative displacement of the temporal waveform maximum obtained from Fig. 4($c_3$) as the black line, and from Fig. 5(a) as the red circles. Note that, the latter was offset along the ps-axis for fitting with the former.

prefer to directly detect the sharp peak of THz spectrum located at $\omega_p$ (in Fig. 2(a)) or the equivalent periodic oscillations in time domain (in Fig. 3(c)) by using a broadband EO crystal. However, both methods might not be practical for the following reasons. Firstly, $\omega_p$ is determined by $N_e$ which varies along the filament (Fig. 3(a)), and thus the overall THz spectral maximum (centered at $\omega_p$) of the entire filament cannot be such a "sharp" peak like that in Fig. 2(a) calculated at a certain $N_e$ value. Normally, a much smooth $\omega_p$ peak is reported in the literature [23, 48, 49]. Secondly, as for the temporal periodic oscillation, it should be mixed with different frequency components because its FT-related $\omega_p$ peak has a considerable bandwidth as mentioned above. Besides, it is well known that the water vapor absorption in air contributes to the random fluctuation of THz amplitude after the main monocycle pulse [66]. These two points would inevitably hinder the expected periodic oscillation from being identified.

## 4 Conclusion

In summary, the current work has provided a new insight into the THz wave generation during single-color femtosecond laser filamentation in gaseous or liquid media with the improved TWA model. On one hand, representative far-field THz radiation profiles observed in experiments of previous publications [6, 8–12, 17, 18], such as (a)symmetrical conical or on-axis THz beam propagation, have been well reproduced with our TWA theory. On the other hand, the plasma electron oscillations have been uncovered as periodic THz emitters along the filament via the effect of THz-plasma resonance, and the temporal advance of THz waveforms is proposed as the practical evidence for the existence of this effect. Owing to the above issues, the underlying nature of THz wave creation by the single-color laser field is approaching to be thoroughly understood in a novel viewpoint of radiated antennas.

Notice that, the suggested TWA perspective is particularly associated with the single-color laser field driven longitudinal photoionization currents. In contrast, a two-color laser driver is commonly understood in terms of transverse currents [67, 68]. However, studies of two- [59, 69] or even multi-color [70–74] laser pumping THz sources, as well as the phased-array plasma filament emitters, might also benefit from the TWA theory. Since it has been discovered that both transverse and longitudinal currents contribute to the THz emission during filamentation [3, 53, 75–81], and the longitudinal currents tend to become dominant within specific ranges

of parameters, like gas pressures, pumping wavelengths, and laser intensities [22, 82, 83]. Therefore, the suggested TWA model is very promising to show its versatility far beyond what has been presented in this work.

## Acknowledgements

The author(s) declare(s) that there is no conflict of interest regarding the publication of this article.

**Author contributions**   J.Z. and Y.P. conceived the idea and designed the experiments. Y.H. Q.W. and Y.C. performed the experiments with help from Z.J. Y.H. and A.P.S. discussed the experimental results.  Simulations were performed by J.Z. and Y.H. All authors wrote the manuscript. Y.Z., S.Z. and L.L. supervised the project.

**Funding information**   This work was supported in part by the Youth Sci-Tech "Qimingx-ing" (22QC1400300) Program of Shanghai, National Natural Science Foundation of China (61988102), 111 Project (D18014), International Joint Lab Program supported by Science and Technology Commission Shanghai Municipality (17590750300), Key project supported by Science and Technology Commission Shanghai Municipality (YDZX20193100004960), and the General Administration of Customs Project (2019hk006,2020hk251). Fundref registry.

## A   Derivation of the THz-plasma resonance effect

According to Fig. 1 of the main text, the process of plasma oscillation can be described as an electron being taken away from the equilibrium position in the post-ionization stage, which could be given by the classical Newton's second law:

$$m_e \frac{\partial v_e}{\partial t} = -eE - m v_e v_e + F_{laser} \, . \tag{7}$$

In Eq. (7), $m_e$ is the mass of an electron with the accelerated speed of $\partial v_e / \partial t$. $eE$ represents the restoring force caused by the attraction from the stationary ion, where $e$ denotes the electronic charge and $E$ is the polarization electric field induced by the electron-ion separation. $m_e v_e v_e$ represents the damping force which is proportional to the momentum $m_e v_e$ and collision frequency $v_e$ of electrons. $F_{laser}$ represents the laser force derived from Ref. [3]:

$$F_{laser} = \frac{e^2}{4c^2 m_e \varepsilon_0 (v_e^2 + \omega_0^2)} \left[ \frac{\partial I}{\partial t} + 2v_e I \right], \tag{8}$$

where $\varepsilon_0$ represents the vacuum permittivity, $\omega_0$ is the laser frequency and $I$ denotes the laser pulse intensity. Flaser mainly consists of the following two effects: the first one, involving the laser intensity gradient $\partial I / \partial t$, is associated with the ponderomotive force; and the second one is proportional to the collision frequency and corresponds to the radiation pressure.

Moreover, as for the conservation of electrons, there is $\partial N_e / \partial t + \partial N_e v_e / \partial z = 0$ in a unit volume as the well-known continuity equation, where $N_e$ is the plasma density. Introducing integration to this equation and then taking $N_{e0} = \delta N_e + N_e$ and $v_e / c \rightarrow 0$ into account, it gives $\delta N_e = N_{e0} v_e / c$ (for $t = 0$, $\delta N_e = 0$ and $v_e = 0$). If the Gauss' law of div$D = e\delta N_e$ is further considered, $v_e = (\varepsilon_0 / e N_{e0}) \times \partial E / \partial t$ can be deduced. This equation of $v_e$ and the plasma fre-

quency $\omega_{pe} = (N_{e0} e^2/m_e \varepsilon_0)^{1/2}$ as well as Eq. (8) were substituted into Eq. (7), which is now in form of

$$\frac{\partial^2 E}{\partial t^2} + \omega_{pe}^2 E + \nu_e \frac{\partial E}{\partial t} = \frac{e\omega_{pe}^2}{4c^2 m_e \varepsilon_0 (\nu_e^2 + \omega_0^2)}[\frac{\partial I}{\partial t} + 2\nu_e I]. \tag{9}$$

Via Fourier transform ($\partial/\partial t \sim -i\omega$), Eq. (9) yields:

$$(-\omega^2 + \omega_{pe}^2 - i\nu_e\omega)E(\omega) = \frac{e\omega_{pe}^2}{4c^2 m_e \varepsilon_0 (\nu_e^2 + \omega_0^2)}[-i\omega + 2\nu_e]I(\omega). \tag{10}$$

In view of the relationship of $j(\omega) = i\omega\varepsilon_0 E(\omega)$ between the displacement current and the polarization electric field, and assuming the laser pulse with duration of $\tau_L$ to be in the sine form of $I(\omega) = I_0 \sin(\omega\tau_L/2)\exp(-i\omega\tau_L/2)/\omega[(\omega\tau_L/2\pi)^2-1]$, the spectral current amplitude ($\omega_0 \gg \nu_e$) can be written as

$$j(\omega) = \varepsilon_0 E_L \omega_{pe} \frac{\omega + 2i\nu_e}{2(\omega^2 - \omega_{pe}^2 + i\nu_e\omega)} \times \frac{\sin(\omega\tau_L/2)}{1-(\omega\tau_L/2\pi)^2} e^{-i\omega\tau_L/2}, \tag{11}$$

where $E_L = e\omega_{pe}I_0/2m_e\varepsilon_0 c^2\omega_0^2$ is the amplitude of the plasma wave driven by the laser ponderomotive force. And the radiated spectrum given by $E_{rad}(\omega)\propto i\omega j(\omega)$ [67] is

$$E_{rad}(\omega) \propto i\omega\varepsilon_0 E_L \omega_{pe} \frac{\omega + 2i\nu_e}{2(\omega^2 - \omega_{pe}^2 + i\nu_e\omega)} \times \frac{\sin(\omega\tau_L/2)}{1-(\omega\tau_L/2\pi)^2} e^{-i\omega\tau_L/2}, \tag{12}$$

which is also shown in the main text as Eq. (3).

## B  The effect of the longitudinal DC electric field on the plasma current

When the plasma filament is applied with a external DC field ($E_{ex}^l$) along the longitudinal axis, Eq. (7) can be transformed into the following Eq. (13):

$$m_e \frac{\partial v_e^l}{\partial t} = -eE_p^l - eE_{ex}^l - m_e v_e^l \nu_e + F_{laser}. \tag{13}$$

Via Fourier transformation, the electron velocity in the longitudinal direction is written as $v_e^l = [e(E_p^l + E_{ex}^l) - F_{laser}]/m_e (i\omega - \nu_e)$, which is further accelerated by the additional $E_{ex}^l$. In this case, the spectral current can be deduced by similar derivation processes as Eq. (7-11), and the result is

$$j(\omega) = \varepsilon_0 E_L \omega_{pe} \frac{\omega + 2i\nu_e}{2(\omega^2 - \omega_{pe}^2 + i\nu_e\omega)} \times \frac{\sin(\omega\tau_L/2)}{[1-(\omega\tau_L/2\pi)^2]} e^{-i\omega\tau_L/2} + \frac{\varepsilon_0 E_{ex}^l \omega_{pe}^2}{\omega^2 - \omega_{pe}^2 + i\nu_e\omega}. \tag{14}$$

Consequently, the spectral current amplitude is the sum of two terms, i.e., $j = j_\omega + j_{ex}^l$, where the first one is the wake current spectrum as shown in Eq. (11), and the second spectral current is excited by the external electric field:

$$j_{ex}^l(\omega) = \frac{\varepsilon_0 E_{ex}^l \omega_{pe}^2}{\omega^2 - \omega_{pe}^2 + i\nu_e\omega}. \tag{15}$$

For comparison between $j_\omega$ and $j_{ex}^l$, calculations of Eq. (11) and (15) have been carried out, and the results are shown in Fig. (6). It can be seen that with $E_{ex}^l$ of 1 kV/cm (green dot and dash line), the additional current $j_{ex}^l$ is already comparable with the original $j_\omega$ (black line) whose laser wakefield amplitude $E_\omega$ ($=E_L 2\nu_e \tau_L \omega/\omega_{pe}$) is reported in the order of magnitude of several hundred V/cm (e.g., 0.2 kV/cm in Ref. [9]). With a larger $E_{ex}^l$ of 5 or 10 kV/cm as used in Ref. [7], the spectral currents can be further increased significantly as expected. In this case, $j_{ex}^l$ in Eq. (15) dominates the THz wave generation, and the direction of $E_{ex}^l$ (i.e., $\pm E_{ex}^l$) won't change the radiated THz intensity since $j_\omega$ is neglectable [8].

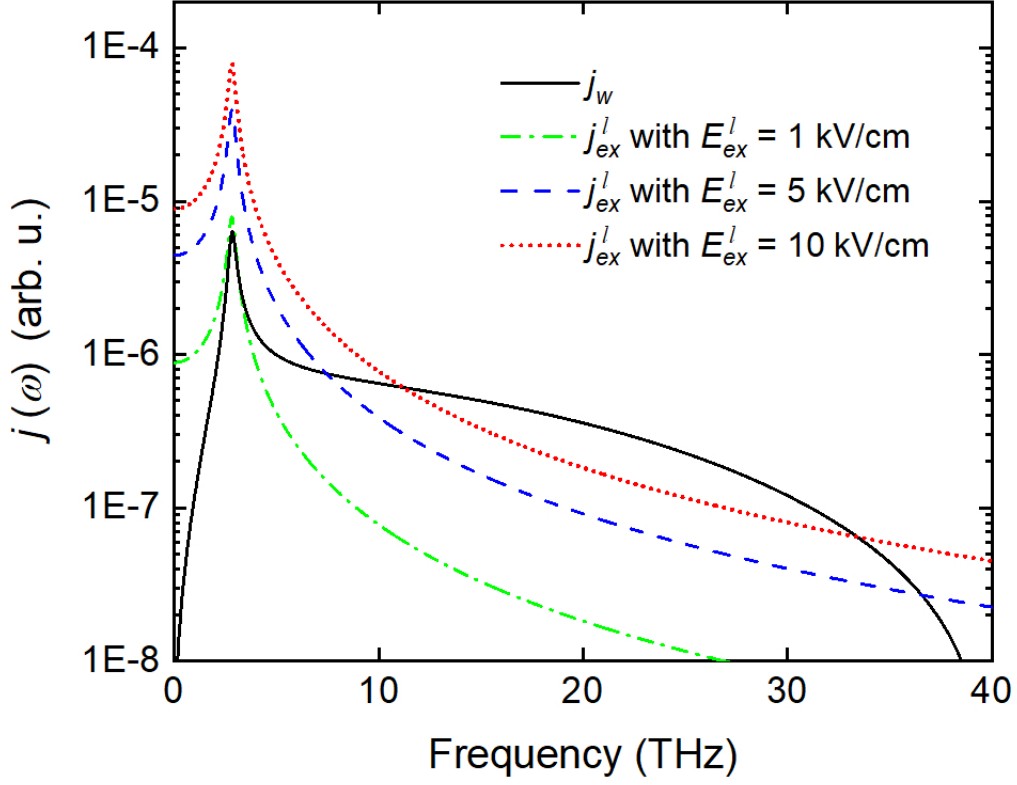

Figure 6: The wake current spectrum (black line) and the additional spectral currents with longitudinal external electric fields of 1 kV/cm (green dot and dash line), 5 kV/cm (blue dashed line) and 10 kV/cm (red dotted line), respectively.

Therefore, we come to the conclusion that the electric current driven by the external field could greatly exceed the current induced by the ponderomotive force, as long as the employed DC value reaches the level of kV/cm. At the same time, not only the plasma current, but also the THz energy is much enhanced during filamentation. In consideration of the above analyses with our microscopic model, we accordingly detailed the definition of the gain coefficient n as $n = 1 + |E_{ex}^l/E_\omega|$ in our macroscopic (TWA) model as shown in Table 1 of the main text.

Revisiting Fig. 6, towards low frequencies such as 0.1 THz, the spectral current $j_{ex}$ (green dot and dash line) excited by the external electric field $E_{ex}$ is about two orders of magnitude larger than the original $j_\omega$ (black line) given by the laser wakefield. For this reason, $j_\omega$ was not included in our calculations at 0.1 THz (Table 1), and the THz yield is solely proportional to $|j_{ex}|$ (also $|E_{ex}|$, see Eq. 15). Thus we stated that the direction of $E_{ex}$ (i.e., $\pm E_{ex}$) won't change the radiated THz intensity. However, if larger THz frequency (>0.1 THz) is considered, one can also see in Fig. 6 that, $j_\omega$ rapidly increased and even exceeded $j_{ex}$ at around 2 THz. This time, $j_\omega$ is no longer negligible and the total THz yield is decided by Eq. 14 as $[j_\omega + j_{ex}(E_{ex})]$, which is now influenced by the sign of $E_{ex}$. From Eq. 14, we can further pre-

dicted that the minimum THz yield is obtained at a negative $E_{ex}$ ($= -j_{\omega}/j_{ex}$), thus in agreement with Fig. 2(a) of Ref. [84].

## C  The effect of the transverse DC electric field on the plasma current

As for the presence of a transverse DC field $E_{ex}^t$, Eq. (7) is re-considered in the transverse direction and can be changed into

$$m_e \frac{\partial v_e^t}{\partial t} = -eE_p^t + eE_{ex}^t - m_e v_e^t v_e.$$
(16)

In the same way as Eq. (7-11), the transverse spectral current amplitude can be obtained:

$$j_{ex}^t(\omega) = -\frac{\varepsilon_0 E_{ex}^t \omega_{pe}^2}{\omega^2 - \omega_{pe}^2 + i v_e \omega},$$
(17)

which has the similar expression with that induced by the longitudinally external field as shown in Eq. (15), and the effect of $j_{ex}^t$ will also be noticeable when $E_{ex}^t$ is close to (or larger than) $E_{\omega}$.

In this condition, the combination of the transverse current $j_{ex}^t$ and the original longitudinal current $j_{\omega}$(Eq. (11)) determines the total THz yield, for which reason $\varphi = \arctan(E_{ex}^t/E_{\omega})$ is defined to describe the angle between the direction of the current element and the filament axis in the TWA model as shown in Table 1 of the main text.

## D  Estimate of the THz conversion efficiency

Here we calculated the THz power conversion efficiency given by the resonance radiation theory. Based on the proportional relationship shown in Eq. (12), it is now transformed into Eq. (18) by being multiplied with a factor of $A/4\pi\varepsilon_0 c^2$ [85], where $A$ is the area of the current cross section, roughly in diameter of 100 $\mu$m in case of a plasma filament.

$$E_{rad}(\omega) = \frac{A}{4\pi\varepsilon_0 c^2} \times i\omega\varepsilon_0 E_L \omega_{pe} \frac{\omega + 2i v_e}{2(\omega^2 - \omega_{pe}^2 + i v_e \omega)} \times \frac{\sin(\omega\tau_L/2)}{1 - (\omega\tau_L/2\pi)^2} e^{-i\omega\tau_L/2}.$$
(18)

Then, the power spectral density of the generated THz wave can be described as

$$\frac{dW_{rad}(\omega)}{d\omega} = \frac{|E_{rad}(\omega)|^2}{2\eta} S,$$
(19)

where $\eta = 377\ \Omega$ is the optical impedance, and $S$ is the area of THz radiation spot, normally within a diameter of millimeter scale (set 2 mm as an example). At last, the generated THz power can be obtained by spectral integral of Eq. (19) in the range from 0 to 200 THz, and the result is about $1.26\times10^{-10}$ W. Reminding ourselves that the input laser power in our case is $\sim$1 W, thus the conversion efficiency is of the order of magnitude of $\sim10^{-10}$. This value is slightly larger than the reported $10^{-11}$ [7] for the transition-Cherenkov mechanism, but still much lower than that of the two-color laser pumping scheme ($\sim10^{-4}$).

# E  Calculation of the electron collision frequency

For laser plasma in ambient air pressure similar with our situation, the mean free time of electron collision (or the scattering time) is constantly reported on the order of hundreds of femtoseconds [46,59,60,86–88], corresponding to the electron collision frequency in terahertz scale (mostly 1-5 THz) [3, 6–8, 43, 44], based on which 2 THz was adopted in the main text (Section 3.1).

We have also revisited the value of electron collision frequency with formulas widely accepted in literature in the community [3,43,44,89–92]. The electron collision frequency ($\nu_e$) being considered during THz wave radiation is given by

$$\nu_e = \nu_{en} + \nu_{ei}, \tag{20}$$

where $\nu_{en}$ and $\nu_{ei}$ are the electron-neutral and electron-ion collision frequencies, respectively. Here, $\nu_{en}$ can be computed by

$$\nu_{en} = N_n \sigma_0 (k_B T_e/m_e)^{1/2}, \tag{21}$$

where $N_n = N_{n0} - N_e$ is the neutral density with the ambient $N_{n0} = 2.7 \times 10^{19}$ cm$^{-3}$, $k_B$ is the Boltzmann constant, and $\sigma_0 \sim 10^{-15}$ cm$^2$ [93–95] is a typical value for the scattering cross section. In practical units, $\nu_{en}$ can be simplified as $\nu_{en}$[Hz]= $4 \times 10^{-8} N_n$[cm$^{-3}$]$T_e$[eV]$^{1/2}$. And $\nu_{ei}$ is given by

$$\nu_{ei} = \frac{\sqrt{2}}{3\sqrt{\pi}} N_e \left(\frac{Ze^2}{4\pi\varepsilon_0}\right)^2 \frac{4\pi}{m_e^{1/2} T_e^{3/2}} ln\Lambda, \tag{22}$$

which can also be written simply as $\nu_{ei}$[Hz]= $3 \times 10^{-6}$ (ln$\Lambda$)$N_e$[cm$^{-3}$]$/T_e$[eV]$^{3/2}$. For the parameters of our experiment, with ln$\Lambda$=5, $N_e$=$10^{17}$ cm$^{-3}$ and $T_e = 1$ eV at 300K, $\nu_e$ is calculated as $\nu_e = 1.05$ ($\nu_{en}$) + $1.45$ ($\nu_{ei}$) = 2.5 THz, which is close to the value used in the main text.

# F  Discussions about the plasma-THz resonance phenomenon

As put in Ref. [88], strong resonant enhancement of the THz radiation was observed if the plasma frequency ($\omega_p/2\pi$) was close to the inverse pulse length of the pumping laser ($t_0 = 120 \sim 140$ fs, thus $1/t_0$ is in THz band). Furthermore, this work indicates that (i) the frequency of THz emission varied with the gas density (by changing the ambient gas pressure) and was close to the bulk plasma frequency as shown in its Fig. 3, and (ii) the THz signal was attributed to the nonlinear current in its Eq. (1), whose denominator contained the item of $[1-(\omega_p/\omega)^2]$. Both (i) and (ii) reveal that maximum THz signals could be achieved at $\omega_p=\omega_{THz}$.

Although the concept of plasma-THz resonance was not directly raised in Ref. [88], the above relationships showed the clue. However, here we want to emphasize solely on the phenomenal similarities between Ref. [88] and our work (Section 3 of the main text, e.g., Fig. 2b$_1$), while the underlying mechanisms might be very different because of the used relativistic laser pulses for generating THz radiation in Ref. [88]. Accordingly. the resulting experimental configurations, like the focused laser intensity or the plasma density, could be much larger than the laser equipment used by us, leading to more complicated physics.

It can also be noticed that Ref. [88] is often cited as the first demonstration of single-color laser-ionization induced THz wave generation in the community, but the following works were mostly carried out with non-relativistic lasers. In the non-relativistic case, we base on the quasi-Cherenkov model (Ref. [6] of the main text) and understand the THz radiation better with the concept of travelling-wave antenna in certain aspects, as we put in the Introduction of the

main text. On the other hand, the plasma filament has been studied in quite a different way in this work. For us, the THz resonant radiation was mainly investigated by its longitudinal evolution along the filament (Fig. 3-5 of the main text), rather than collecting the signal from the whole filament. This is a new view in our field to the best of our knowledge.

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
