# Peer review of "Traveling-wave antenna model for terahertz radiation from laser-plasma interactions"

_SciPost Physics, doi:SciPost Phys. Core 5, 046 (2022)_

## Round 1 · Referee Report · Anonymous (Referee 1) · 2022-4-19

Strengths

1-A comprehensive model is proposed for photo-ionized THz generation in air.
2-A good (re)visit to classic and new phenomena in experiments in this field.

Weaknesses

1-Revision is need. See the Requested changes.

Report

In this manuscript, the traveling-wave antenna (TWA) model has been proposed for interpreting the underlying mechanism of THz wave generation from a single-color filament in gas or liquid. Diverse far-field THz radiation patterns from different experimental configurations were well reproduced by the TWA model, proving the applicability of this theory. Moreover, the proposed model was also studied in a microscopic point of view, and the THz resonance radiation effect was theoretically and experimentally presented as another crucial consideration when dealing with the plasma filament based THz source.

In summary, the combination of concepts of TWA and resonance radiation is new and novel in the field, and the corresponding interpretation is rigorous. The manuscript is well organized and written by providing comprehensive studies, including detailed theoretical analyses and experimental demonstrations. The results and discussions are scientifically interesting to readers in related communities. Therefore, I can recommend this work for publication in Scipost Physics, as long as the following comments are properly replied.

Requested changes

1-In Section 2, with a longitudinal DC field E_ex, the authors investigated the 0.1 THz radiation from the filament, and it was stated that the direction of E_ex won’t change the radiated THz intensity. I notice that, recently, similar studies at larger THz frequencies was reported [PRE 102, 063211 (2020)], and the THz yield was found to be affected by the sign of E_ex. Could the authors’ model explain the results (at >0.1 THz) in the literature? This would help validate the suggested theory in this work.

2-In Section 3, the THz resonance effect (spectral peak) was demonstrated by the temporal advance of THz waveforms from length-controlled plasma filaments as shown in Fig. 4 and 5. I think this approach is clever. On the other hand, the spectral peak shift with a varying N_e was reported in [PRL 71(17), 2725 (1993), Fig. 3]. Is this another potential evidence of the THz-plasma resonance effect? The authors should discuss this point in the manuscript.

3-In the Supplemental Material, as far as I understand, Section B and C are relatively crucial since they play the role of bridging the macroscopic TWA model and the microscopic model, according to the last paragraphs of these two sections. If I were correct, this bridging effect should be clearly presented in the main text to strength the connection between Section 2 and 3.

  • validity: high
  • significance: high
  • originality: top
  • clarity: top
  • formatting: excellent
  • grammar: excellent

Author:  Yuchen Hui  on 2022-05-02  [id 2425]

(in reply to Report 1 on 2022-04-19)

1-Thank the reviewer for pointing this issue out. As shown in Fig. S1 of our manuscript, towards low frequencies such as 0.1 THz, the spectral current $j_{ex}$ (green dot and dash line) excited by the external electric field $E_{ex}$ is about two orders of magnitude larger than the original one $j_{w}$ (black line) given by the laser wakefield. For this reason, $j_{w}$ was not included in our calculations at 0.1 THz, and the THz yield is solely proportional to |$j_{ex}$| (also |$E_{ex}$|, see Eq. S9). Thus we stated that the direction of $E_{ex}$ (i.e., ±$E_{ex}$) won’t change the radiated THz intensity.

However, if larger THz frequency (>0.1 THz) is considered, one can also see in Fig. S1 that, $j_{w}$ rapidly increased and even exceeded $j_{ex}$ at around 2 THz. This time, $j_{w}$ is no longer negligible and the total THz yield is decided by Eq. S8 as [$j_{w}$+$j_{ex}$($E_{ex}$)], which is now influenced by the sign of $E_{ex}$. From Eq. S8, we can further predicted that the minimum THz yield is obtained at a negative $E_{ex}$ (=-$j_{w}$/$j_{ex}$), thus in agreement with Fig. 2(a) of Ref. [Phys. Rev. E 102, 063211 (2020)].

2-Thanks for the reviewer’s information. As put in Ref. [Phys. Rev. Lett. 71(17), 2725 (1993)], strong resonant enhancement of the THz radiation was observed if the plasma frequency $(\omega_p/2\pi)$ was close to the inverse pulse length of the pumping laser ($t_0$ = 120~140 fs, thus 1/$t_0$ is in THz band). Although the concept of plasma-THz resonance was not directly raised in this work, the above relationship showed the clue.

Furthermore, the PRL work indicates that (i) the frequency of the THz emission varies with gas density (by changing the ambient gas pressure) and is close to the bulk plasma frequency as shown in its Fig. 3, and (ii) the THz signal was attributed to the nonlinear current in its Eq. (1), whose denominator contains the item of $[1-(\omega_p/\omega)^2]$. Both (i) and (ii) reveal that maximum THz signals could be achieved at $\omega_p=\omega_{THz}$ (i.e., resonance effect).

Hence, we think the PRL work might share several phenomenological conclusions with our Section 3 (e.g., Fig. 2b$_1$), while the plasma filament is studied in quite different ways. For us, the THz resonant radiation is mainly investigated by its longitudinal evolution along the filament (Fig. 3-5), rather than collecting the signal from the whole filament. This is a new view in the community, to the best of our knowledge.

3-The reviewer is right that Section B and C of the Supplemental Material are informative, since they formed the gain coefficient ($n$) and the rotation angle factor ($\phi$) from the microscopic point of view, respectively, which were then used in the macroscopic TWA model (Section 2.2 and 2.3).

Following the reviewer’s suggestion, we are going to emphasize this point in the first part of Section 3 to strength its link with Section 2.

Anonymous on 2022-06-05  [id 2556]

(in reply to Yuchen Hui on 2022-05-02 [id 2425])

Sorry for my late reply. I am happy to see the authors' responses, and all my concerns have been resolved. I have also learned from the following two comments (id 2466, 2528). Now I can recommend publication of this work in Scipost Physics.

Anonymous on 2022-05-15  [id 2466]

(in reply to Yuchen Hui on 2022-05-02 [id 2425])

In the above response 2, it is interesting to see discussions on [PRL 71, 2725 (1993)]. I keep in mind of this outstanding work because of the used relativistic laser pulses for generating THz radiation. And the resulting experimental configurations, like the focused laser intensity or the plasma density, could be much larger than the laser equipment used by the authors. For this, I think the authors should be very careful on the statements associated with this PRL work. I don’t deny some observations are similar with the reviewed work, but the underlying mechanism might be very different.

Anonymous on 2022-05-26  [id 2528]

(in reply to Anonymous Comment on 2022-05-15 [id 2466])

Thank the reviewer for the advice. We emphasized on the phenomenal similarities between [PRL 71, 2725 (1993)] and our work, as we mentioned in the above Response #2, and we have the same thought with the reviewer that the relativistic laser filamentation brings more complicated physics than ours. Although this PRL work is often cited as the first demonstration of single-color laser-ionization induced THz wave generation in our field, the following works are mostly carried out with non-relativistic lasers. In the non-relativistic case, we base on the quasi-Cherenkov model (Ref. 6 of the manuscript) and understand the THz radiation better with the concept of travelling-wave antenna in certain aspects, as we put in the Introduction of the manuscript.

---

## Round 1 · Referee Report · Anonymous (Referee 2) · 2022-4-21

Strengths

This work offers a versatile electrical-to-optical model by treating the laser plasma filament as an antenna.

Weaknesses

Please see the report below.

Report

This work presents the traveling-wave antenna (TWA) model for explaining the important THz wave creation by the photo-ionization in air. The novelty of this work can be partly judged from the very recent publication by Prof. A. M. Zheltikov in Optics Letters [OL 46, 4984-4987 (2021)], which has also revealed the superiority of the antenna theory for reviewing previous experimental facts and expediting future studies in this field. Compared with the OL letter, this research article has provided much more detailed theoretical and experimental works, proving the modal validity on many distinct conditions (e.g. four tables in the manuscript).

For the above reasons, I can recommend publication of this work. Here are my questions that the authors should take good care of before acceptance: 1. I don’t like the statement of “universal” before the TWA model, because every theory has its own preconditions and application range. I think “versatile” might be more suitable for describing the model. 2. Does the water film absorption of THz wave need to be considered in the model (in Table IV), and why? 3. The plasma frequency wp, and wp/sqrt(2), wp/sqrt(3) are frequently observed in the literature according to the plasma geometry. Which one fits the proposed model best during the THz-filament interaction?

Requested changes

Please see the report. Especially the last two points.

  • validity: top
  • significance: high
  • originality: high
  • clarity: top
  • formatting: excellent
  • grammar: excellent

Author:  Yuchen Hui  on 2022-04-24  [id 2411]

(in reply to Report 2 on 2022-04-21)

1-Thanks for the reviewer’s advice. “universal” will be removed or replaced all over the revised manuscript.

2-According to Ref. [17] cited in Table IV, the THz intensity after the water film absorption is given by $I_{THz} = I_0exp[-\alpha d/2cos\theta_t]$, where $\alpha = 100 cm^{-1}$ is the power absorption coefficient [DOI: 10.1109/ICIMW.2009.5325641], d = 120 $\mu$m is the thickness of the water film, and $\theta_t$ is the THz incidence angle on the water-to-air interface, as described in Table IV. On this interface, THz waves can’t be coupled out if its total internal reflection occurs, and the critical angle can be easily calculated as 25 degree ($n_{THz}$ = 2.33 in water). Thus $\theta_t$ varies between -25 and +25 degree, and $I_{THz}$ is between 0.51$I_0$ and 0.54$I_0$.

Due to this little change of $I_{THz}(\theta_t)$, the water absorption is not considered in our work when profiling the THz radiation distribution from the water film (Table IV) for the sake of convenience.

3-The reviewer is correct that $\omega_p$, $\omega_p/\sqrt2$ and $\omega_p/\sqrt3$ are all characteristic frequencies related with the plasma. Normally, they correspond to the plasma frequencies of the plasma line, cylinder and sphere, respectively. In our case, we considered mostly the plasma as a filament, whose length is longer than 10 mm (e.g., in Table I) while its column diameter is much smaller as about 50 $\mu$m. Thus we chose $\omega_p$ in the derivation of the model (Section 3).

On the other hand, if the plasma cylinder and $\omega_p/\sqrt2$ were allowed, the main conclusion of this work (THz radiation profiles induced by the TWA model and THz-plasma resonance) won’t be changed, because $\omega_p/\sqrt2$ is still in THz band, and the THz-plasma interaction remains.

At last, $\omega_p/\sqrt3$ is not our case since neither our laser beam is tightly focused, nor a micro-plasma-sphere [Optica 2(4) 366-369 (2015)] is studied in this work.

Anonymous on 2022-05-15  [id 2465]

(in reply to Yuchen Hui on 2022-04-24 [id 2411])

I’m satisfied with the response.

Please feel free with “universal” if the authors have enough confidence on the proposed model.

Response 2 and 3 are also needed in the revised paper.

Anonymous on 2022-05-26  [id 2529]

(in reply to Anonymous Comment on 2022-05-15 [id 2465])

Thank you for your comment.

---

## Round 2 · Author Response

We gratefully appreciate the precious time that the editor has spent on our manuscript, and also the constructive comments from the reviewers, which are significant to promote the quality of our research.

---

## Round 2 · List of Changes

1-“universal” has been removed or replaced all over the revised manuscript.
2-The reasons for neglecting the water film absorption of THz waves have been addressed (in blue) in the third paragraph of Section 2.8 of the main text.
3-The choice of plasma frequency dependent on our experimental situation is explained (in blue) in the paragraph before Section 3.2 of the main text.
4-Discussions about the influence of the external electrical field on the THz yield are addressed (in blue) in the last paragraph of Section B in the Supporting Materials.
5-We have emphasized on the bridging effect of Supplemental Materials A-C in the first paragraph of Section 3.1 of the main text (in blue) in order to strength the connection between Section 2 and 3.
6-Discussions about the plasma-THz resonance phenomenon have been added in the paragraph before Section 3.2 of the main text, and also in the revised Supplemental Materials as Section F.

---

## Editorial Decision

published